# Effect of Optimal Alcohol-Based Hand Rub among Nurse Students Compared with Everyday Practice among Random Adults; Can Water-Based Hand Rub Combined with a Hand Dryer Machine Be an Alternative to Remove *E. coli* Contamination from Hands?

**DOI:** 10.3390/microorganisms11020325

**Published:** 2023-01-28

**Authors:** Hans Johan Breidablik, Lene Johannessen, John Roger Andersen, Hilde Søreide, Ole T. Kleiven

**Affiliations:** 1Center of Health Research, Western Norway University of Applied Sciences and Helse Førde HF, 6812 Førde, Norway; 2Thelma Indoor Air & Working Environment, Department Microbiology, 7037 Trondheim, Norway; 3Department of Health and Caring Sciences, Western Norway University of Applied Sciences, 6812 Førde, Norway

**Keywords:** hand hygiene, disinfection, alcohol-based hand rub, water-based hand rub, hand dryer, bacteria

## Abstract

Efficient hand hygiene is essential for preventing the transmission of microorganisms. Alcohol-based hand rub (ABHR) is a recommended method. We compared health personnel (skilled nurse students) with random adults to study the effect of an ABHR procedure. A water-based hand rub (WBHR) procedure, using running tap water and a hand-drying machine, was also investigated. The study included 27 nurse students and 26 random adults. Hands were contaminated with *Escherichia coli*, and concentrations of colony forming units (CFU/mL) were determined before and after ABHR or WBHR. Concentrations after ABHR were 1537 CFU/mL (nurse students) and 13,508 CFU/mL (random adults) (*p* < 0.001). One-third of participants reported skin irritation from daily ABHR. Concentrations after WBHR were 41 CFU/mL (nurse students) and 115 CFU/mL (random adults) (*p* < 0.011). The majority of participants (88.5%) preferred the WBHR method. Results from 50 air samples from filtered air from the hand dryer outlet showed no CFU in 47 samples. A significant difference between the two groups was shown for the ABHR method, indicating that training skills are important for efficient hand hygiene. Surprisingly, the WBHR method seemed to have a significant effect in largely removing transient bacteria from hands.

## 1. Introduction

Preventing the spread of virulent microorganisms is essential for infection control programs, with hand disinfection playing a pivotal role [1]. Well-known measures during the COVID-19 pandemic, as well as everyday routines to prevent the spreading of virulent virus or antibiotic resistant bacterial pathogens in hospitals, include frequent hand disinfection with alcohol-based hand sanitizers. Alcohol-based hand rub (ABHR) is the universally recommended hand disinfection method, and the World Health Organization guidelines state that disinfection with alcohol effectively will eradicate transient bacteria and is skin-friendly [1,2]. One of the advantages of ABHR is that dispensers can be easily provided in public places and in the proximity of patients [1,2].

The microbiological flora on the hands can largely be classified into resident and transient groups [3]. Bacteria in the normal flora group are stable and reproduce locally. They are generally non-virulent, and concentrations can only be reduced with disinfection. The normal flora bacteria also play a protective role in the skin barrier as part of the innate immune system [4,5]. By contrast, transient microbes do not usually reproduce locally while on hands, and they are usually not viable for longer periods of time. Transient bacteria can be highly virulent and pathogenic, such as the ESKAPE bacteria, and they are easily transmitted to infect or colonize skin wounds or in dermatitis [1,6].

Healthcare workers (HCW) have a high prevalence of skin irritations associated with the frequent hand sanitization at work in hospitals, which is contrary to the perception of ABHR being skin-friendly [7,8,9]. Consequently, and despite the protocols for hand sanitization at hospitals, the adherence to hand hygiene protocols in general and among HCWs ranges from as low as 5% up to 81% [10].

The widespread use of ABHR among the general population has also brought the question of adverse skin symptoms into focus [11]. Given the increase in hand disinfection in the general population, an important question is whether ABHR is equally effective when practiced among the general population, as compared to when practiced among skilled HCW. Additionally, it is important to examine whether random adults in the general public also experience unpleasant skin symptoms, and whether efficient alternative methods for hand disinfection are required.

In two earlier studies, we concluded that a water-based hand rub (WBHR), using ozonized tap water or soap water, might be more effective than the ABHR practice to remove transient *Escherichia coli* from artificially contaminated hands. This was also the preferred method among the nurse students who composed the study cohort [12,13].

A WBHR method needs to be followed by hand drying. Open warm-air dryers are associated with the possible spread of microorganisms into the surrounding air, and therefore, they are increasingly being removed from many public areas [14,15]. For example, Margas et al. [16] reported a risk of cross-contamination from washrooms to the environment, and Ma [14] tested restroom hand dryers at retail outlets and detected both bacterial and fungal colonies. Best et al. [15] reported that the open hand-drying method carries the risk of airborne bacteria dissemination in real-world settings. For these reasons, hands dryers have been replaced by the use of paper towels, which has led to an increased paper towel consumption and the creation of unwanted waste [17]. A survey among US adults assessed the self-reported hand drying practices in public bathrooms and found an increased preference for electric hand dryers, wiping hands on cloths or even shaking of hands for drying, and a lower preference for paper towels during the COVID-19 pandemic [18].

In the present study, we tested a WBHR method using only temperate tap water along with a closed low-pressure hand dryer machine (2022/01, SMCL001EP; Smart Cleaner Ltd., Nordfjordeid, Norway). In this hand dryer machine, the air around the hand drying area is constantly maintained at a negative or low pressure. The air circulating inside the machine is exposed to antimicrobial UVC-light and passes through filters before the exhaust outlet at the bottom of the machine.

By comparing ABHR with WBHR/hand drying method for disinfection, the present study seeks to answer the following questions:How effective is ABHR when practiced optimally among skilled HCW (nurse students) compared to the everyday practice among random adults?Can a WBHR/hand drying method be an alternative, without the risk of spreading bacteria into the surrounding air?What are the participant preferences regarding the two methods?

## 2. Materials and Methods

The test method was modified from European Standard EN 1500:2013 [19] which specifies test requirements to simulate practical conditions for establishing whether a hygienic hand product reduces the release of transient microorganism when rubbed onto the artificially contaminated hands of volunteers. We modified this standard method to enable us to investigate transmittable and viable *E. coli* only, regardless of the amount and composition of the resident skin flora and other particulate artefacts present on the skin surface. We used bacterial samples taken from hands before and after a hygienic treatment to investigate differences in the concentrations of the traceable *E. coli* bacteria.

In addition, we modified the standard test method in order to simulate an everyday clinical situation by including HCW (nurse students) as test subjects, as well as volunteers from workplaces in the nearby geographic area kindly asked to participate. Records of everyday routines for hand washing and preferences for hand hygiene methods constituted an important part of this study.

An important modification included using 3 mL (2 × 1.5 mL) of 85% (vol/vol) alcohol disinfectant as a reference, instead of 6 mL (2 × 3 mL) of 60% (vol/vol), as specified by the standard. Another important modification was spreading liquid samples of bacteria onto selective agar plates by using calibrated 10 µL loops, rather than preparing sample dilutions before spreading volumes of 0.1 mL or 1.0 mL of liquid samples onto nonselective agar plates. Consequently, we were able to avoid the use of time-consuming spectrophotometry for measuring light absorption in solutions to roughly estimate nonspecific particulate concentrations in several dilution steps, as specified by the standard. Instead, the initial bacterial liquid solution used to contaminate all of the hands was prepared with an estimated bacterial concentration >10^9^ CFU/mL. By reducing time-consuming steps, we managed to execute the test procedure in a very time-efficient manner, which is of importance when working with rapidly proliferating bacterial cells in cultures.

Finally, instead of 12 to 15 test subjects, as specified in the standard, we included two test groups consisting of 27 skilled nursing students (group A) and 26 adult volunteers recruited from workplaces in nearby geographic locations (group B). There was no further formal randomization of the test subjects. Some characteristics of the test groups are shown in Table 1. All participants had healthy skin, without cuts or abrasions, and no visible signs of dermatitis on their hands during the tests. All participants were above 18 years of age.

### 2.1. Preparation of the Bacterial Culture and Contamination of Hands

The American Type Culture Collection (ATCC) strain 25922 (*Escherichia coli*) was used as the testing organism. This strain originates from normal flora and is internationally recognized as being a nonpathogenic group 1 organism; therefore, it is specifically chosen to meet health and safety requirements in experimental studies. The test organism recommended in the standard is an *E. coli* strain available from the National Collection of Type Cultures (NCTC 10538). However, this strain and the ATCC strain are essentially identical and thus equally suitable for the purpose of this study.

ATCC 25922 was pre-cultivated for 18 to 24 h, at 37 ± 1 °C, on selective agar plates (MacConkey No. 3, Thermo Fisher Scientific, Waltham, MA, USA). Thereafter, 1 colony was picked and further spread on nonselective tryptic soy agar (Thermo Fisher Scientific), followed by incubation for 18–24 h, at 37 ± 1 °C. A single colony from the tryptic soy agar plate was inoculated into 10 mL of sterile tryptic soy broth (TSB) (Thermo Fisher Scientific) before another cultivation, and finally, the 10 mL bacterial solution was added to a total volume of 1 L of TSB, which was then cultivated for 18 to 24 h to make a cloudy bacterial solution estimated to have a final concentration of >10^9^ CFU/mL. The 1 L bacterial solution was poured into two 500 mL glass containers, and all participants dipped their hands up to the mid-metacarpals in turn for 5 s. This solution was made to contaminate the hands of all test subjects on testing day 1, and a freshly made solution was used on testing day 2.

### 2.2. Test Procedure after Contamination of Hands

After contamination of hands and air drying for 3–4 min, the test subjects rubbed their fingertips on the base of a Petri dish containing 10 mL of sterile TSB, with a separate Petri dish for each hand. Then, approximately 1 mL of each liquid sample was transferred into an Eppendorf microtube (Thermo Fisher Scientific), and the tubes were brought in a cooling bag to the microbiological laboratory within 24 h.

On day 1, groups A and B used 85% Antibac Hand Rub (ABHR) as a reference disinfectant. Antibac consists of ethanol supplemented with 2-propanol (http://www.antibac.no (accessed on 24 January 2023). Nursing students (group A) used 2 doses of ABHR (2 × 1.5 mL) from a dispenser (see Figure 1). They were instructed and observed by a hygiene nurse to perform the hand rubbing procedure optimally for 30 s. Participants in group B (random adults from nearby workplaces) were instructed to use the ABHR as they normally would in an everyday setting. They did this individually, in separate rooms, and they were not instructed or observed by a hygiene nurse. Following the ABHR procedure, post-samples were collected from hands after 3–4 min of air drying and by rubbing fingertips on the base of a Petri dish containing 10 mL of sterile TSB using separate Petri dishes for each hand. Approximately 1 mL of each post-sample was transferred into an Eppendorf microtube (Thermo Fisher Scientific) and brought to the laboratory together with the pre-samples.

Two days later, on testing day 2, the same test participants performed a water-based hand rub (WBHR) procedure. After contamination of hands and pre-sampling as already described, they washed their hands in temperate running tap water for 30 s followed by drying in a closed hand dryer machine for 30 seconds (Figure 2).

Post-samples were collected and brought to the laboratory as previously described. During the 30 s of active hand drying, the exhaust air from the hand dryer machine was sampled using an Oxoid Air Sampler (Thermo Fisher Scientific) with laminar airflow (50 L/30 s) directed onto the surface of a 90 mm agar plate (MacConkey No. 3, Thermo Fisher Scientific). These air sample agar plates were incubated for 18 to 24 h, at 37 ± 1 °C.

### 2.3. Cultivation of Liquid Samples on Agar Plates and Calculation of CFU Concentrations

In the laboratory, microtubes containing liquid samples were thoroughly mixed in a vortex mixer before 10 µL calibrated loops were used to transfer liquid samples onto parallel MacConkey agar plates (Thermo Fisher Scientific). Liquid was extensively spread on the agar plate before incubation for 18 to 24 h at 37 ± 1 °C. After incubation, all CFU per plate were counted (<300 CFU per plate), and the concentration (CFU/mL) was calculated by multiplying the colony counts with the dilution factor of 100. We used the arithmetic mean of 2 plate counts for each hand sample to obtain the pre- and post-sample values for further statistical analysis. CFU counts >300/plate were not applicable, and concentrations were noted >30,000 CFU/mL.

### 2.4. Statistical Analyses

Population characteristics are reported as means, standard deviations, medians, ranges, and raw numbers and percentages. The differences between groups A and B were analyzed using an independent *t*-test, the Fisher exact test, or the Mann–Whitney U-test. In a secondary analysis, the Spearman rank test was used to examine the relationship between factors such as age, gender, disinfectant concentration, and the *E. coli* count. We calculated two-sided *p*-values as continuous probability indicators. IBM SPSS Statistics version 27 was used for statistical analysis, and Microsoft Excel 2016 was used to make graphics.

### 2.5. Ethics

The participants were provided written and oral information about the study and were thereafter invited to participate. All the participants who were included provided written consent. The study was approved by the Regional Committee for Medical Research Ethics and the Norwegian Data Inspectorate (approval number: 263147).

## 3. Results

Table 1 presents the number, age, and gender of participants in groups A and B. Group A comprised 27 nurse students and was significantly younger (age 21 to 35 years) and had more female participants (74.4%) than group B, which had 26 participants (random adults) who were 20 to 80 years old and 53.8% female. Group A participants were each instructed to use two doses of alcohol disinfectant (approximately 3 mL) from the hygiene nurse instructor, while participants in group B were told to use the same dose as in their everyday ABHR practice. More than half of participants in group B (53.8%) used only one dose, while the rest used two doses (except one who used 3 doses). The ABHR dispenser is shown in Figure 1.

### 3.1. The Pre-Disinfection Tests on Day 1 (ABHR)

The pre-disinfection tests showed an *E. coli* concentration of >30,000 CFU/mL (upper cutoff) on 46 out of 54 hands in group A, and on 48 out of 52 hands in group B.

### 3.2. The Post-Disinfection Tests on Day 1 (ABHR)

The post-disinfection tests conducted after using ABHR showed that mean and median concentration of *E. coli* for both hands was 1537 (SD 2361) and 800 CFU/mL in group A (nursing students). In group B (random adults), the mean and median was 13,508 (SD 14,803) and 5450 CFU/mL.

For the right hand, the mean (median) concentration was 819 (300) in group A and 9073 (2500) in group B. The corresponding values for the left hand were 719 (400) for group A and 4435 (3000) for group B. All of these differences were highly significant (*p* < 0.001) (Table 2).

For group A, 4 of the 27 participants had no remaining *E. coli* colonies on their hands after ABHR, while none of the 26 participants in group B achieved this. Figure 3 illustrates the mean concentrations (CFU/mL) for each hand after the ABHR procedure in group A and group B.

### 3.3. The Pre-Disinfection Tests on Day 2 (WBHR)

When the WBHR method combined with active hand drying was used, the pre-disinfection concentrations were generally lower in both groups compared with that of day 1 concentrations. Mean (median) concentrations for both hands were 17,011 (13,300) CFU/mL in group A and 12,446 (8900) in group B. Table 3 shows the corresponding values for right and left hands.

### 3.4. The Post-Disinfection Tests on Day 2 (WBHR)

With the combined method of WBHR and the closed hand dryer machine, the remaining mean (median) *E. coli* concentration on both hands was 41 (0) CFU/mL for group A and 115 (100) CFU/mL for group B (*p* = 0.011). For the right hand, the concentration was 15 (0) CFU/mL for group A and 65 (0) CFU/mL for group B (*p* = 0.036). For the left hand, 26 (0) CFU/mL for group A and 50 (0) CFU/mL for group B (*p* = 0.226), as also illustrated in Figure 4.

Further, 16 of the 27 participants in group A and 9 of the 26 participants of group B had no viable *E. coli* colonies left on their hands.

In group B, there was a significant age-related difference in the remaining *E. coli* colonies for ABHR (*p* = 0.003) and WBHR/hand dryer (*p* = 0.029).

### 3.5. Hand Dryer Machine and Bacterial Contamination in the Air Surrounding

The closed hand dryer machine is illustrated in Figure 4. This directs a warm air flow over the hands, with a negative pressure inside the device to prevent aerosols from the hand drying process escaping from the machine and into the surrounding air. The drying air flow that passes through a charcoal filter before the outlet was tested for the potential content of *E. coli* colonies. Among 50 air samples from beneath the dryer machine, 47 samples were clean, and only 1 *E. coli* colony was detected in each of the remaining 3 air samples.

### 3.6. User Preferences and Adverse Skin Symptoms

In both groups, a majority of the participants preferred the WBHR method (A; 74% and B; 96%).

Both groups were asked about adverse skin symptoms from regular daily use of ABHR disinfection earlier, but without specifying the actual number of procedures per day. Overall, 30% of A and 31% of B reported this, but for group B, only slight symptoms were noted, while among the nurse students, some participants also reported heavier symptoms (see Table 4).

A majority (A, 100%; B, 97%) found that the hand dryer machine was efficient and convenient to dry their hands. Furthermore, 74% in group A and 85% in group B thought that the machine could or should replace the use of paper towels after washing of hands.

### 3.7. Age and Gender

Correlation analysis for gender and post-ABHR *E. coli* colonies on the hands showed to be non-significant. With regard to age, this was 0.597 for right hand (*p* < 0.000), and 0.446 for left hand (*p* = 0.001) (Spearman).

## 4. Discussion

### 4.1. ABHR Effectiveness among Skilled Nurse Students Compared to Random Adults

In the present study, ABHR could not eradicate all viable *E. coli* colonies from heavily contaminated hands, even with an optimal hand disinfection procedure among nurse students. The efficiency was even lower when practiced in an everyday setting among random adult participants. None of the participants in group B achieved *E. coli*-free hands in this study. The importance of education and training skills to perform ABHR correctly is well known from the literature [20,21]. We conducted two earlier studies with the ABHR procedures and found similar results among nurse students [12,13]. It has been demonstrated in a study that up to 86% of test subjects in a real world setting use only small doses of alcohol disinfectant, if not instructed otherwise, as low as approximately 0.75 mL [22]. In the present study, 46% (12 out of 26 participants) in group B used at least two doses (3 mL) from the dispenser, as recommended by the manufacturer. Based on these findings, we suggest that a practice of not using the sufficient dose of alcohol disinfectant to cover both of the hands may challenge the effectiveness of hand hygiene procedures in the general public.

### 4.2. WBHR Followed by an Active Hand Dryer Procedure as an Alternative Method

The participants rubbed their *E. coli*-contaminated hands with only running tap water (without soap) (WBHR) followed by the use of a hand dryer machine. Surprisingly, the results from post-sampling of hands showed that concentrations of *E. coli* were low. The difference between groups A and B was also moderate when this method was used, and we found that the difference in concentrations (CFU/mL) between the right and left hands of random adults were also quite low. Based on the results from our earlier study, when test subjects used a water and soap combination [13], we did not expect that the removal of soap could have resulted in a low *E. coli* concentration after the WBHR. The rubbing of hands in running water is probably the main reason why transient bacteria can be removed from the hands, although the addition of soap would have an additional effect as a solvent of fatty cell membranes of bacteria cells.

A disadvantage of the WBHR method is that it is highly dependent on access to clean and running water. Such access is limited or absent in some areas of the world. The use of hand dryers in general is also dependent on electricity.

### 4.3. Bacterial Spread from the Hand Dryer Machine to the Air Surrounding

We tested for possible contamination of air surrounding the closed hand dryer machine by sampling the air at the outlet of the machine. In 47 out of the 50 collected air samples, no *E. coli* colonies were seen. Only 1 CFU was seen in each of the remaining three air samples. These findings indicate that a closed hand dryer can be regarded as safe in public settings and that bacteria are not spread to the surrounding air.

User evaluation of this hand dryer machine was also generally positive. The majority of participants experienced an efficient and sufficient hand drying process, and most of the participants expressed a positive view with regard to a replacement of paper towels with a closed hand dryer machine in public areas. This would avoid the excessive waste created by paper towel use.

### 4.4. Adverse Skin Effects and Participant Preferences

Adverse skin effects related to ABHR are well known among HCW, and work-related skin diseases are estimated to affect at least 20% of professionals [23]. For some HCW, skin problems lead to regular dermatitis, making clinical work problematic. We have reported in an earlier study that approximately 30% of participants experienced skin problems such as dryness, burning and redness because of the frequent ABHR [2]. In the present study, participants expressed that they experienced adverse skin symptoms such as dryness and burning from a frequent ABHR use, with more serious symptoms among some of the nurse students.

Age seemed to have an effect on the hand cleaning outcome, as well as the dose of alcohol disinfectant. Suen et al. [24] found that females and middle age were parameters associated with better hand hygiene, which might explain the age-dependent effect observed in our study. Skin changes that naturally occur with age may also explain these findings. Laube [25] reported that elderly individuals have increased susceptibility to skin infections due to age-related anatomical, physiological, and environmental factors, and Skowron et al. [26] reported that the topography of the skin associated with the ageing process (e.g., wrinkle formation) could influence the skin microbiome composition. Finally, hand size has shown to be of importance for the effective coverage of alcohol disinfectants and eradication of bacteria on hands [27], which could indicate that the used volume of disinfectant is of vital importance.

### 4.5. Strengths and Weaknesses of This Study

The strength of this study is that the two different groups of participants (skilled/unskilled) completed both days of testing and that the test results could be compared. The random adult group consisted of a broad spectrum of ages and backgrounds, while the skilled nurse students represented a more homogeneous group. The results are limited by the small number of participants in each group. Additionally, the low pre-disinfection concentrations (contaminated hands) on testing day 2 may have affected the results regarding the significant differences found in ABHR compared to WBHR/hand dryer.

Our primary intention for this study was to compare the effect of an ABHR method to a WBHR method; however, the low pre-test concentration of *E. coli* on the hands on testing day 2 did not allow us to answer to this. The bacterial culture was kept alive and cultivated identically before each of the testing days, so we do not have a good explanation for this phenomenon.

Further, the combined use of WBHR with the hand dryer made it impossible to evaluate the contributions of these two parameters separately. However, earlier studies have shown that different forms of WBHR (e.g., with soap or ozonized water) are equally or more effective than ABHR alone [13]. We therefore suggest that running water over the hands can be an essential part of the cleaning of hands in general.

## 5. Conclusions

Our study suggests that ABHR has a significantly different antibacterial effect when it is used by skilled (nurse students) compared to unskilled users (random adults). Knowledge of the correct use is probably the most important finding in this study. When considering the WBHR, it seems to effectively and largely remove transient *E. coli* colonies on both hands in both the skilled and unskilled group of participants. Due to discrepancy in the pre-disinfection bacterial concentrations, we cannot conclude that WBHR is more/less effective than ABHR. A hand dryer machine was well accepted by both groups, and the majority of test subjects preferred this drying method instead of using, e.g., paper towels.

Considering hand disinfection as the most essential measure to prevent the spreading of virulent microorganisms in institutions and in public places, we conclude that the choice of hand hygiene method should be considered carefully, both regarding the risk of skin irritations and hand hygiene effectiveness.

## Figures and Tables

**Figure 1 microorganisms-11-00325-f001:**
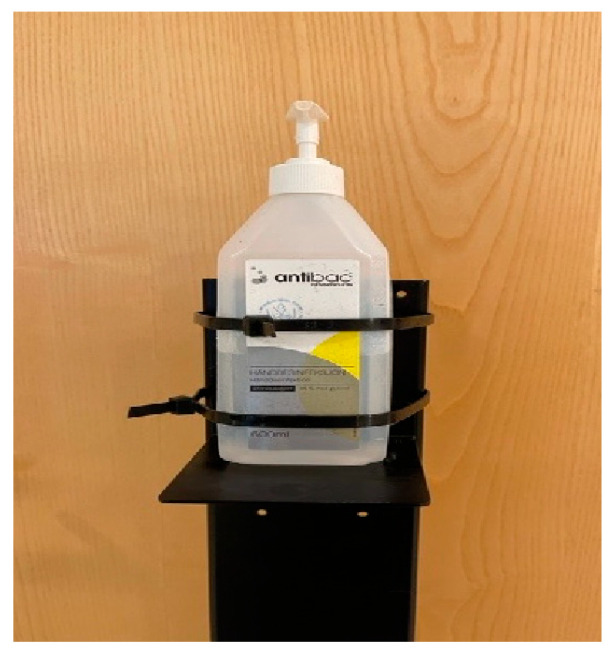
ABHR dispenser (Antibac) used by the participants.

**Figure 2 microorganisms-11-00325-f002:**
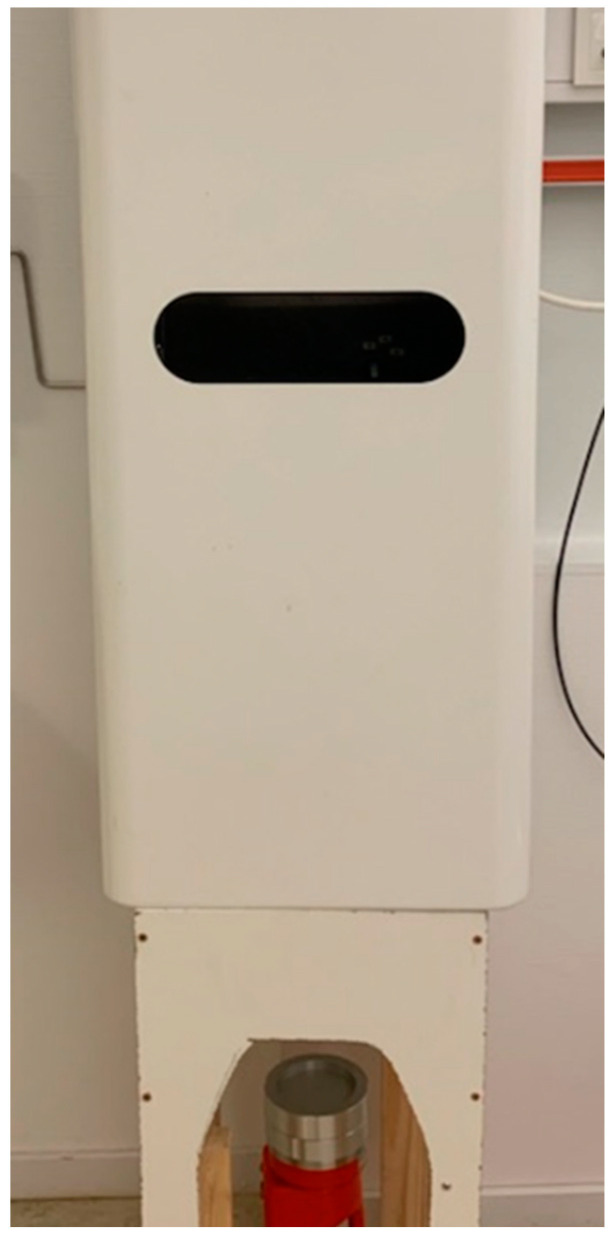
The closed hand dryer machine used in the study. The Oxoid Air Sampler is placed at air outlet from the device.

**Figure 3 microorganisms-11-00325-f003:**
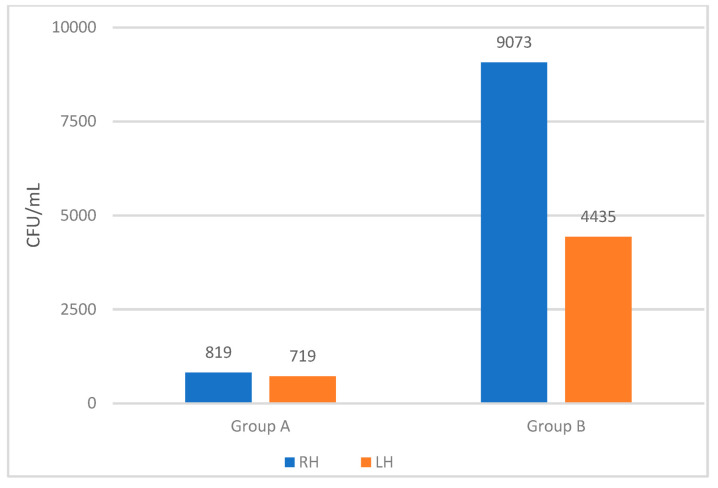
Mean concentrations (CFU/mL) for the two hands after using ABHR for group A (nurse students) and group B (random adults). LH: left hand, RH: right hand.

**Figure 4 microorganisms-11-00325-f004:**
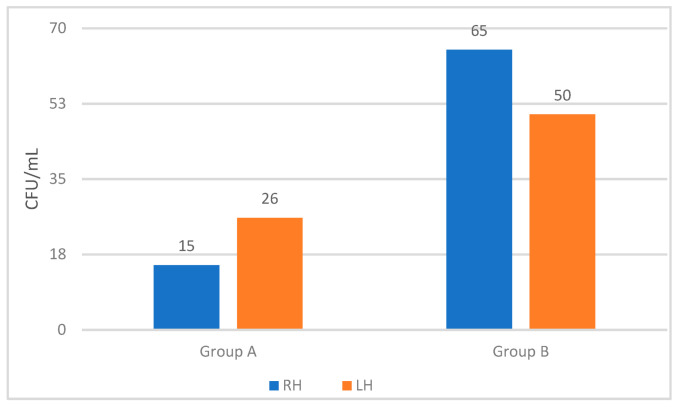
Mean concentrations (CFU/mL) for the two hands after using WBHR and hand dryer machine for group A (nurse students) and group B (random adults). LH: left hand, RH: right hand.

**Table 1 microorganisms-11-00325-t001:** Characteristics of the two groups (A and B).

Variables	A (n = 27) (Nurse Students)	B (n = 26) (Random Adults)	*p*-Values
Age (years) mean (SD) (min–max)	24.4 (3.7) (21–35)	45.3 (16.7) (20–80)	<0.001 ^a^
Gender, women, n	20 (74.4)	14 (53.8)	0.158 ^b^
Number of Antibac doses used, n (%)			<0.001 ^b^
1	0 (0)	14 (53.8)	
2	27 (100)	11 (42.3)	
3	0 (0)	1 (3.8)	

^a^ Mann–Whitney U-test; ^b^ Fisher exact test.

**Table 2 microorganisms-11-00325-t002:** Post-test concentrations (CFU/mL) after ABHR (day 1). All pre-tests showed concentrations of >30,000 CFU/mL.

Variables	A (n = 27) (Nurse Students)	B (n = 26) (Random Adults)	*p*-Values
Both hands			<0.001 ^a^
Mean (SD)	1537 (2361)	13,508 (14,803)	
Median	800 (10,200)	5450	
Right hand			<0.001 ^a^
Mean (SD)	819 (1620)	9073 (11,961)	
Median	300	2500	
Left hand			<0.001 ^a^
Mean (SD)	719 (1072)	4435 (5154)	
Median	400	3000	

Note: ^a^ Mann–Whitney U-test.

**Table 3 microorganisms-11-00325-t003:** Pre- and post-concentrations (CFU/mL) on testing day 2, before and after using WBHR and a hand dryer machine for group A (nurse students) and group B (random adults).

Variables	A (n = 27)	B (n = 26)	*p*-Values
Both hands (pre-test)			
Mean (SD)	17,011 (15,358)	12,446 (11,293)	
Median	13,300	8900	
Both hands (post-test)			0.011 ^a^
Mean (SD)	41 (50)	115 (116)	
Median	0	100	
Right hand (pre-test)			
Mean (SD)	7952 (8805)	7539 (7663)	
Median (range)	4000	5050	
Right hand (post-test)			0.036 ^a^
Mean (SD)	15 (36)	65 (102)	
Median	0	0	
Left hand (pre-test)			
Mean (SD)	9059 (9015)	4908 (5978)	
Median	5700	2900	
Left hand (post-test)			0.226 ^a^
Mean (SD)	26 (45)	50 (76)	
Median	0	0	

^a^ Mann–Whitney U-test for group differences in post-values.

**Table 4 microorganisms-11-00325-t004:** Self-reported variables and opinions from the participants (number/percentage).

N (%)		Group A (Nurse Students)	Group B (Random Adults)
Skin symptoms with earlier regular ABHR use	Little	5 (19)	8 (31)
	Some	3 (11)	0 (0)
	Dermatitis	1 (4)	0 (0)
Preferred method for hand disinfection	ABHR (Alcohol-based hand rub)	7 (26)	1 (4)
	WBHR (Water-based hand rub)/hand dryer	20 (74)	25 (96)
Drying effect of the hand dryer	Very good	11 (41)	16 (62)
	Good	16 (59)	9 (35)
	Insufficient	0 (0)	1 (4)
The dryer can substitute paper towel use	Yes	20 (74)	22 (85)
	Uncertain	3 (11)	0(0)
	No	4 (15)	4 (15)
The preferred method for hand disinfection	ABHR (Alcohol-based hand rub)	7 (26)	1 (4)
	WBHR (Water-based hand rub)	20 (74)	25 (96)

## Data Availability

The dataset in this study is not publicly available as the participant consent and approval from the Regional Committee for Medical Research Ethics prevent sharing individual data in public repositories. However, the data will be available from the corresponding author upon reasonable request.

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
