# Peer review of "Effect of Optimal Alcohol-Based Hand Rub among Nurse Students Compared with Everyday Practice among Random Adults; Can Water-Based Hand Rub Combined with a Hand Dryer Machine Be an Alternative to Remove E. coli Contamination from Hands?"

_microorganisms, 2023, doi:10.3390/microorganisms11020325_

Round 1
Reviewer 1 Report
The manuscript titled “Effect of optimal alcohol-based hand rub among nurse students compared with everyday practice among random adults. Can water-based hand rub combined with a hand dryer be an alternative?” authored by Breidablik et al. is an interesting study that investigates the effective methods of hand sanitization in a clinical setting and the general public. The beginning of the Introduction section talks about the ongoing COVID-19 pandemic and the need of the hand sanitization which gives the initial impression to the reader that the study would be on an antiviral hand sanitizer, which is not the case. Rather, the authors have used strains of E. coli to examine the effectiveness of their methods. Therefore, I would suggest the authors mention briefly about the ESKAPE pathogens in the clinical settings and the need of a sanitizer that can effectively disinfect the hands of the nurses or clinicians to prevent the infections or contamination with ESKAPE pathogens. I believe that would be a more appropriate background for this study. While the authors used only E. coli and not all the ESKAPE pathogens, it would still justify this study as a pilot study for one of these bacterial pathogens.
I would suggest the authors make few changes to the manuscript as below:
tL58: Do not start a paragraph with the word ‘however’.
L63: ‘HCWs’. Spell out at the first use.
L65: ‘health care workers’ can be abbreviated here, if spelled out in line 63.
L65-66: “In particular, preserving lipids, fatty acids, and resident microbial flora is important”. Please provide more background why this is important.
L70: Replace the words “these individuals” with ‘general public’, if that is what the authors meant.
L71: Replace the words “exist” with ‘are required’. The current statement sounds like if something exists contemporarily and needed to be discovered, which is not the case here. The new methods can be developed if deemed necessary, as per requirement.
L72: “In two earlier studies”.
L77: Delete ‘However”.
L277-279: After how many days (two days?) or how frequent use of the sanitizers resulted adverse skin symptoms in study participants? Did the symptoms get worse with repeated and consistent use of the sanitizer? What is the probability of the adverse skin symptoms after long-term daily use? Please explain briefly for clarity.
L303-305: The authors hypothesize that the rubbing time may be too short, or the volume used may be insufficient. What are the recommended parameters for effectiveness? Will a higher volume and longer rubbing time influence or improve the effectiveness of disinfection? Please explain clearly.
Line 308: WBRH?
Line 315: it might also be suitable for the children.
L319-320: Paragraph is too short and is out of context. Should be deleted.
Line 389-390: Authors concluded that WBHR could be effective method for children. This is when children were not among the participants as all the participants were above 18 years of age. This statement is misleading and should be deleted.
Other comments:
1. This reviewer believes that the title might be restructured for inclusion of E. coli because the study was based on that one pathogen only.
2. Please give a brief background of the ESKAPE pathogens in the Introduction section. Sentence on COVID-19 can be removed, which to this reviewer appears bit misleading in the beginning, as if the methods were tested against SARS-CoV2.
3. This reviewer believes that there is enough scope of improving on the Introduction section.
4. ABHR does not appear to be effective disinfectant for nursing students. This is missing from the conclusion.
5. Conclusion section is weak and has a misleading statement. It needs to be re-written with recommendations.
6. With authors reporting two earlier studies from their group on hand sanitization, the present study might have used various combinations of approaches to find the best method in clinical and nonclinical settings. The study design has a scope of improvement. The present study is a brief investigation of only two different approaches, and therefore might be considered as a ‘Brief Report’ rather than a full article.
Author Response
Reviewer 1
Comments and Suggestions for Authors
- The manuscript titled “Effect of optimal alcohol-based hand rub among nurse students compared with everyday practice among random adults. Can water-based hand rub combined with a hand dryer be an alternative?” authored by Breidablik et al. is an interesting study that investigates the effective methods of hand sanitization in a clinical setting and the general public. The beginning of the Introduction section talks about the ongoing COVID-19 pandemic and the need of the hand sanitization which gives the initial impression to the reader that the study would be on an antiviral hand sanitizer, which is not the case. Rather, the authors have used strains of E. coli to examine the effectiveness of their methods. Therefore, I would suggest the authors mention briefly about the ESKAPE pathogens in the clinical settings and the need of a sanitizer that can effectively disinfect the hands of the nurses or clinicians to prevent the infections or contamination with ESKAPE pathogens. I believe that would be a more appropriate background for this study. While the authors used only E. coli and not all the ESKAPE pathogens, it would still justify this study as a pilot study for one of these bacterial pathogens.
Thanks. We have edited the introduction according to this feedback.
- tL58: Do not start a paragraph with the word ‘however’.
Corrected.
- L63: ‘HCWs’. Spell out at the first use.
Corrected.
- L65: ‘health care workers’ can be abbreviated here, if spelled out in line 63.
Corrected.
- L65-66: “In particular, preserving lipids, fatty acids, and resident microbial flora is important”. Please provide more background why this is important.
We have rewritten this section for clarity.
- L70: Replace the words “these individuals” with ‘general public’, if that is what the authors meant.
Corrected.
- L71: Replace the words “exist” with ‘are required’. The current statement sounds like if something exists contemporarily and needed to be discovered, which is not the case here. The new methods can be developed if deemed necessary, as per requirement.
Corrected.
- L72: “In two earlier studies”.
Corrected.
- L77: Delete ‘However”.
Corrected.
- L277-279: After how many days (two days?) or how frequent use of the sanitizers resulted adverse skin symptoms in study participants? Did the symptoms get worse with repeated and consistent use of the sanitizer? What is the probability of the adverse skin symptoms after long-term daily use? Please explain briefly for clarity.
“Few participants reported dermatitis from regular ABHR use (only 1 in group A), but 30% of group A reported slight or some skin symptoms, while 31% in group B reported only slight skin symptoms.”
This section is rewritten for clarity:
Both groups were asked about adverse skin symptoms from regular daily use of ABHR disinfection earlier, but without specifying the actual number of procedures per day. 30% of A and 31% of B reported this, but for group B only slight symptoms were noted while among the nurse students also some participants reported heavier symptoms (see table 4).
- L303-305: The authors hypothesize that the rubbing time may be too short, or the volume used may be insufficient. What are the recommended parameters for effectiveness? Will a higher volume and longer rubbing time influence or improve the effectiveness of disinfection? Please explain clearly.
- Line 308: WBRH?
WBHR. Corrected.
- Line 315: it might also be suitable for the children.
It could be suitable. But as we comment, it needs to be evaluated.
L319-320: Paragraph is too short and is out of context. Should be deleted.
Corrected.
- Line 389-390: Authors concluded that WBHR could be effective method for children. This is when children were not among the participants as all the participants were above 18 years of age. This statement is misleading and should be deleted.
Corrected.
- This reviewer believes that the title might be restructured for inclusion of E. coli because the study was based on that one pathogen only.
Corrected.
- Please give a brief background of the ESKAPE pathogens in the Introduction section. Sentence on COVID-19 can be removed, which to this reviewer appears bit misleading in the beginning, as if the methods were tested against SARS-CoV2. This reviewer believes that there is enough scope of improving on the Introduction section.
We have edited this section accordingly.
- ABHR does not appear to be effective disinfectant for nursing students. This is missing from the conclusion.
This could be the true. However, due to discrepancy in the pre-disinfection bacterial concentrations we think its best to be careful when commenting differences in effects between WBHR and ABHR. However, we have edited the conclusion for clarity.
- Conclusion section is weak and has a misleading statement. It needs to be re-written with recommendations.
We have rewritten this section for clarity.
Reviewer 1
Comments and Suggestions for Authors
- The manuscript titled “Effect of optimal alcohol-based hand rub among nurse students compared with everyday practice among random adults. Can water-based hand rub combined with a hand dryer be an alternative?” authored by Breidablik et al. is an interesting study that investigates the effective methods of hand sanitization in a clinical setting and the general public. The beginning of the Introduction section talks about the ongoing COVID-19 pandemic and the need of the hand sanitization which gives the initial impression to the reader that the study would be on an antiviral hand sanitizer, which is not the case. Rather, the authors have used strains of E. coli to examine the effectiveness of their methods. Therefore, I would suggest the authors mention briefly about the ESKAPE pathogens in the clinical settings and the need of a sanitizer that can effectively disinfect the hands of the nurses or clinicians to prevent the infections or contamination with ESKAPE pathogens. I believe that would be a more appropriate background for this study. While the authors used only E. coli and not all the ESKAPE pathogens, it would still justify this study as a pilot study for one of these bacterial pathogens.
Thanks. We have edited the introduction according to this feedback.
- tL58: Do not start a paragraph with the word ‘however’.
Corrected.
- L63: ‘HCWs’. Spell out at the first use.
Corrected.
- L65: ‘health care workers’ can be abbreviated here, if spelled out in line 63.
Corrected.
- L65-66: “In particular, preserving lipids, fatty acids, and resident microbial flora is important”. Please provide more background why this is important.
We have rewritten this section for clarity.
- L70: Replace the words “these individuals” with ‘general public’, if that is what the authors meant.
Corrected.
- L71: Replace the words “exist” with ‘are required’. The current statement sounds like if something exists contemporarily and needed to be discovered, which is not the case here. The new methods can be developed if deemed necessary, as per requirement.
Corrected.
- L72: “In two earlier studies”.
Corrected.
- L77: Delete ‘However”.
Corrected.
- L277-279: After how many days (two days?) or how frequent use of the sanitizers resulted adverse skin symptoms in study participants? Did the symptoms get worse with repeated and consistent use of the sanitizer? What is the probability of the adverse skin symptoms after long-term daily use? Please explain briefly for clarity.
“Few participants reported dermatitis from regular ABHR use (only 1 in group A), but 30% of group A reported slight or some skin symptoms, while 31% in group B reported only slight skin symptoms.”
This section is rewritten for clarity:
Both groups were asked about adverse skin symptoms from regular daily use of ABHR disinfection earlier, but without specifying the actual number of procedures per day. 30% of A and 31% of B reported this, but for group B only slight symptoms were noted while among the nurse students also some participants reported heavier symptoms (see table 4).
- L303-305: The authors hypothesize that the rubbing time may be too short, or the volume used may be insufficient. What are the recommended parameters for effectiveness? Will a higher volume and longer rubbing time influence or improve the effectiveness of disinfection? Please explain clearly.
- Line 308: WBRH?
WBHR. Corrected.
- Line 315: it might also be suitable for the children.
It could be suitable. But as we comment, it needs to be evaluated.
L319-320: Paragraph is too short and is out of context. Should be deleted.
Corrected.
- Line 389-390: Authors concluded that WBHR could be effective method for children. This is when children were not among the participants as all the participants were above 18 years of age. This statement is misleading and should be deleted.
Corrected.
- This reviewer believes that the title might be restructured for inclusion of E. coli because the study was based on that one pathogen only.
Corrected.
- Please give a brief background of the ESKAPE pathogens in the Introduction section. Sentence on COVID-19 can be removed, which to this reviewer appears bit misleading in the beginning, as if the methods were tested against SARS-CoV2. This reviewer believes that there is enough scope of improving on the Introduction section.
We have edited this section accordingly.
- ABHR does not appear to be effective disinfectant for nursing students. This is missing from the conclusion.
This could be the true. However, due to discrepancy in the pre-disinfection bacterial concentrations we think its best to be careful when commenting differences in effects between WBHR and ABHR. However, we have edited the conclusion for clarity.
- Conclusion section is weak and has a misleading statement. It needs to be re-written with recommendations.
We have rewritten this section for clarity.
Reviewer 1
Comments and Suggestions for Authors
- The manuscript titled “Effect of optimal alcohol-based hand rub among nurse students compared with everyday practice among random adults. Can water-based hand rub combined with a hand dryer be an alternative?” authored by Breidablik et al. is an interesting study that investigates the effective methods of hand sanitization in a clinical setting and the general public. The beginning of the Introduction section talks about the ongoing COVID-19 pandemic and the need of the hand sanitization which gives the initial impression to the reader that the study would be on an antiviral hand sanitizer, which is not the case. Rather, the authors have used strains of E. coli to examine the effectiveness of their methods. Therefore, I would suggest the authors mention briefly about the ESKAPE pathogens in the clinical settings and the need of a sanitizer that can effectively disinfect the hands of the nurses or clinicians to prevent the infections or contamination with ESKAPE pathogens. I believe that would be a more appropriate background for this study. While the authors used only E. coli and not all the ESKAPE pathogens, it would still justify this study as a pilot study for one of these bacterial pathogens.
Thanks. We have edited the introduction according to this feedback.
- tL58: Do not start a paragraph with the word ‘however’.
Corrected.
- L63: ‘HCWs’. Spell out at the first use.
Corrected.
- L65: ‘health care workers’ can be abbreviated here, if spelled out in line 63.
Corrected.
- L65-66: “In particular, preserving lipids, fatty acids, and resident microbial flora is important”. Please provide more background why this is important.
We have rewritten this section for clarity.
- L70: Replace the words “these individuals” with ‘general public’, if that is what the authors meant.
Corrected.
- L71: Replace the words “exist” with ‘are required’. The current statement sounds like if something exists contemporarily and needed to be discovered, which is not the case here. The new methods can be developed if deemed necessary, as per requirement.
Corrected.
- L72: “In two earlier studies”.
Corrected.
- L77: Delete ‘However”.
Corrected.
- L277-279: After how many days (two days?) or how frequent use of the sanitizers resulted adverse skin symptoms in study participants? Did the symptoms get worse with repeated and consistent use of the sanitizer? What is the probability of the adverse skin symptoms after long-term daily use? Please explain briefly for clarity.
“Few participants reported dermatitis from regular ABHR use (only 1 in group A), but 30% of group A reported slight or some skin symptoms, while 31% in group B reported only slight skin symptoms.”
This section is rewritten for clarity:
Both groups were asked about adverse skin symptoms from regular daily use of ABHR disinfection earlier, but without specifying the actual number of procedures per day. 30% of A and 31% of B reported this, but for group B only slight symptoms were noted while among the nurse students also some participants reported heavier symptoms (see table 4).
- L303-305: The authors hypothesize that the rubbing time may be too short, or the volume used may be insufficient. What are the recommended parameters for effectiveness? Will a higher volume and longer rubbing time influence or improve the effectiveness of disinfection? Please explain clearly.
- Line 308: WBRH?
WBHR. Corrected.
- Line 315: it might also be suitable for the children.
It could be suitable. But as we comment, it needs to be evaluated.
L319-320: Paragraph is too short and is out of context. Should be deleted.
Corrected.
- Line 389-390: Authors concluded that WBHR could be effective method for children. This is when children were not among the participants as all the participants were above 18 years of age. This statement is misleading and should be deleted.
Corrected.
- This reviewer believes that the title might be restructured for inclusion of E. coli because the study was based on that one pathogen only.
Corrected.
- Please give a brief background of the ESKAPE pathogens in the Introduction section. Sentence on COVID-19 can be removed, which to this reviewer appears bit misleading in the beginning, as if the methods were tested against SARS-CoV2. This reviewer believes that there is enough scope of improving on the Introduction section.
We have edited this section accordingly.
- ABHR does not appear to be effective disinfectant for nursing students. This is missing from the conclusion.
This could be the true. However, due to discrepancy in the pre-disinfection bacterial concentrations we think its best to be careful when commenting differences in effects between WBHR and ABHR. However, we have edited the conclusion for clarity.
- Conclusion section is weak and has a misleading statement. It needs to be re-written with recommendations.
We have rewritten this section for clarity.
Reviewer 1
Comments and Suggestions for Authors
- The manuscript titled “Effect of optimal alcohol-based hand rub among nurse students compared with everyday practice among random adults. Can water-based hand rub combined with a hand dryer be an alternative?” authored by Breidablik et al. is an interesting study that investigates the effective methods of hand sanitization in a clinical setting and the general public. The beginning of the Introduction section talks about the ongoing COVID-19 pandemic and the need of the hand sanitization which gives the initial impression to the reader that the study would be on an antiviral hand sanitizer, which is not the case. Rather, the authors have used strains of E. coli to examine the effectiveness of their methods. Therefore, I would suggest the authors mention briefly about the ESKAPE pathogens in the clinical settings and the need of a sanitizer that can effectively disinfect the hands of the nurses or clinicians to prevent the infections or contamination with ESKAPE pathogens. I believe that would be a more appropriate background for this study. While the authors used only E. coli and not all the ESKAPE pathogens, it would still justify this study as a pilot study for one of these bacterial pathogens.
Thanks. We have edited the introduction according to this feedback.
- tL58: Do not start a paragraph with the word ‘however’.
Corrected.
- L63: ‘HCWs’. Spell out at the first use.
Corrected.
- L65: ‘health care workers’ can be abbreviated here, if spelled out in line 63.
Corrected.
- L65-66: “In particular, preserving lipids, fatty acids, and resident microbial flora is important”. Please provide more background why this is important.
We have rewritten this section for clarity.
- L70: Replace the words “these individuals” with ‘general public’, if that is what the authors meant.
Corrected.
- L71: Replace the words “exist” with ‘are required’. The current statement sounds like if something exists contemporarily and needed to be discovered, which is not the case here. The new methods can be developed if deemed necessary, as per requirement.
Corrected.
- L72: “In two earlier studies”.
Corrected.
- L77: Delete ‘However”.
Corrected.
- L277-279: After how many days (two days?) or how frequent use of the sanitizers resulted adverse skin symptoms in study participants? Did the symptoms get worse with repeated and consistent use of the sanitizer? What is the probability of the adverse skin symptoms after long-term daily use? Please explain briefly for clarity.
“Few participants reported dermatitis from regular ABHR use (only 1 in group A), but 30% of group A reported slight or some skin symptoms, while 31% in group B reported only slight skin symptoms.”
This section is rewritten for clarity:
Both groups were asked about adverse skin symptoms from regular daily use of ABHR disinfection earlier, but without specifying the actual number of procedures per day. 30% of A and 31% of B reported this, but for group B only slight symptoms were noted while among the nurse students also some participants reported heavier symptoms (see table 4).
- L303-305: The authors hypothesize that the rubbing time may be too short, or the volume used may be insufficient. What are the recommended parameters for effectiveness? Will a higher volume and longer rubbing time influence or improve the effectiveness of disinfection? Please explain clearly.
- Line 308: WBRH?
WBHR. Corrected.
- Line 315: it might also be suitable for the children.
It could be suitable. But as we comment, it needs to be evaluated.
L319-320: Paragraph is too short and is out of context. Should be deleted.
Corrected.
- Line 389-390: Authors concluded that WBHR could be effective method for children. This is when children were not among the participants as all the participants were above 18 years of age. This statement is misleading and should be deleted.
Corrected.
- This reviewer believes that the title might be restructured for inclusion of E. coli because the study was based on that one pathogen only.
Corrected.
- Please give a brief background of the ESKAPE pathogens in the Introduction section. Sentence on COVID-19 can be removed, which to this reviewer appears bit misleading in the beginning, as if the methods were tested against SARS-CoV2. This reviewer believes that there is enough scope of improving on the Introduction section.
We have edited this section accordingly.
- ABHR does not appear to be effective disinfectant for nursing students. This is missing from the conclusion.
This could be the true. However, due to discrepancy in the pre-disinfection bacterial concentrations we think its best to be careful when commenting differences in effects between WBHR and ABHR. However, we have edited the conclusion for clarity.
- Conclusion section is weak and has a misleading statement. It needs to be re-written with recommendations.
We have rewritten this section for clarity.
Reviewer 1
Comments and Suggestions for Authors
- The manuscript titled “Effect of optimal alcohol-based hand rub among nurse students compared with everyday practice among random adults. Can water-based hand rub combined with a hand dryer be an alternative?” authored by Breidablik et al. is an interesting study that investigates the effective methods of hand sanitization in a clinical setting and the general public. The beginning of the Introduction section talks about the ongoing COVID-19 pandemic and the need of the hand sanitization which gives the initial impression to the reader that the study would be on an antiviral hand sanitizer, which is not the case. Rather, the authors have used strains of E. coli to examine the effectiveness of their methods. Therefore, I would suggest the authors mention briefly about the ESKAPE pathogens in the clinical settings and the need of a sanitizer that can effectively disinfect the hands of the nurses or clinicians to prevent the infections or contamination with ESKAPE pathogens. I believe that would be a more appropriate background for this study. While the authors used only E. coli and not all the ESKAPE pathogens, it would still justify this study as a pilot study for one of these bacterial pathogens.
Thanks. We have edited the introduction according to this feedback.
- tL58: Do not start a paragraph with the word ‘however’.
Corrected.
- L63: ‘HCWs’. Spell out at the first use.
Corrected.
- L65: ‘health care workers’ can be abbreviated here, if spelled out in line 63.
Corrected.
- L65-66: “In particular, preserving lipids, fatty acids, and resident microbial flora is important”. Please provide more background why this is important.
We have rewritten this section for clarity.
- L70: Replace the words “these individuals” with ‘general public’, if that is what the authors meant.
Corrected.
- L71: Replace the words “exist” with ‘are required’. The current statement sounds like if something exists contemporarily and needed to be discovered, which is not the case here. The new methods can be developed if deemed necessary, as per requirement.
Corrected.
- L72: “In two earlier studies”.
Corrected.
- L77: Delete ‘However”.
Corrected.
- L277-279: After how many days (two days?) or how frequent use of the sanitizers resulted adverse skin symptoms in study participants? Did the symptoms get worse with repeated and consistent use of the sanitizer? What is the probability of the adverse skin symptoms after long-term daily use? Please explain briefly for clarity.
“Few participants reported dermatitis from regular ABHR use (only 1 in group A), but 30% of group A reported slight or some skin symptoms, while 31% in group B reported only slight skin symptoms.”
This section is rewritten for clarity:
Both groups were asked about adverse skin symptoms from regular daily use of ABHR disinfection earlier, but without specifying the actual number of procedures per day. 30% of A and 31% of B reported this, but for group B only slight symptoms were noted while among the nurse students also some participants reported heavier symptoms (see table 4).
- L303-305: The authors hypothesize that the rubbing time may be too short, or the volume used may be insufficient. What are the recommended parameters for effectiveness? Will a higher volume and longer rubbing time influence or improve the effectiveness of disinfection? Please explain clearly.
- Line 308: WBRH?
WBHR. Corrected.
- Line 315: it might also be suitable for the children.
It could be suitable. But as we comment, it needs to be evaluated.
L319-320: Paragraph is too short and is out of context. Should be deleted.
Corrected.
- Line 389-390: Authors concluded that WBHR could be effective method for children. This is when children were not among the participants as all the participants were above 18 years of age. This statement is misleading and should be deleted.
Corrected.
- This reviewer believes that the title might be restructured for inclusion of E. coli because the study was based on that one pathogen only.
Corrected.
- Please give a brief background of the ESKAPE pathogens in the Introduction section. Sentence on COVID-19 can be removed, which to this reviewer appears bit misleading in the beginning, as if the methods were tested against SARS-CoV2. This reviewer believes that there is enough scope of improving on the Introduction section.
We have edited this section accordingly.
- ABHR does not appear to be effective disinfectant for nursing students. This is missing from the conclusion.
This could be the true. However, due to discrepancy in the pre-disinfection bacterial concentrations we think its best to be careful when commenting differences in effects between WBHR and ABHR. However, we have edited the conclusion for clarity.
- Conclusion section is weak and has a misleading statement. It needs to be re-written with recommendations.
We have rewritten this section for clarity.
Reviewer 1
Comments and Suggestions for Authors
- The manuscript titled “Effect of optimal alcohol-based hand rub among nurse students compared with everyday practice among random adults. Can water-based hand rub combined with a hand dryer be an alternative?” authored by Breidablik et al. is an interesting study that investigates the effective methods of hand sanitization in a clinical setting and the general public. The beginning of the Introduction section talks about the ongoing COVID-19 pandemic and the need of the hand sanitization which gives the initial impression to the reader that the study would be on an antiviral hand sanitizer, which is not the case. Rather, the authors have used strains of E. coli to examine the effectiveness of their methods. Therefore, I would suggest the authors mention briefly about the ESKAPE pathogens in the clinical settings and the need of a sanitizer that can effectively disinfect the hands of the nurses or clinicians to prevent the infections or contamination with ESKAPE pathogens. I believe that would be a more appropriate background for this study. While the authors used only E. coli and not all the ESKAPE pathogens, it would still justify this study as a pilot study for one of these bacterial pathogens.
Thanks. We have edited the introduction according to this feedback.
- tL58: Do not start a paragraph with the word ‘however’.
Corrected.
- L63: ‘HCWs’. Spell out at the first use.
Corrected.
- L65: ‘health care workers’ can be abbreviated here, if spelled out in line 63.
Corrected.
- L65-66: “In particular, preserving lipids, fatty acids, and resident microbial flora is important”. Please provide more background why this is important.
We have rewritten this section for clarity.
- L70: Replace the words “these individuals” with ‘general public’, if that is what the authors meant.
Corrected.
- L71: Replace the words “exist” with ‘are required’. The current statement sounds like if something exists contemporarily and needed to be discovered, which is not the case here. The new methods can be developed if deemed necessary, as per requirement.
Corrected.
- L72: “In two earlier studies”.
Corrected.
- L77: Delete ‘However”.
Corrected.
- L277-279: After how many days (two days?) or how frequent use of the sanitizers resulted adverse skin symptoms in study participants? Did the symptoms get worse with repeated and consistent use of the sanitizer? What is the probability of the adverse skin symptoms after long-term daily use? Please explain briefly for clarity.
“Few participants reported dermatitis from regular ABHR use (only 1 in group A), but 30% of group A reported slight or some skin symptoms, while 31% in group B reported only slight skin symptoms.”
This section is rewritten for clarity:
Both groups were asked about adverse skin symptoms from regular daily use of ABHR disinfection earlier, but without specifying the actual number of procedures per day. 30% of A and 31% of B reported this, but for group B only slight symptoms were noted while among the nurse students also some participants reported heavier symptoms (see table 4).
- L303-305: The authors hypothesize that the rubbing time may be too short, or the volume used may be insufficient. What are the recommended parameters for effectiveness? Will a higher volume and longer rubbing time influence or improve the effectiveness of disinfection? Please explain clearly.
- Line 308: WBRH?
WBHR. Corrected.
- Line 315: it might also be suitable for the children.
It could be suitable. But as we comment, it needs to be evaluated.
L319-320: Paragraph is too short and is out of context. Should be deleted.
Corrected.
- Line 389-390: Authors concluded that WBHR could be effective method for children. This is when children were not among the participants as all the participants were above 18 years of age. This statement is misleading and should be deleted.
Corrected.
- This reviewer believes that the title might be restructured for inclusion of E. coli because the study was based on that one pathogen only.
Corrected.
- Please give a brief background of the ESKAPE pathogens in the Introduction section. Sentence on COVID-19 can be removed, which to this reviewer appears bit misleading in the beginning, as if the methods were tested against SARS-CoV2. This reviewer believes that there is enough scope of improving on the Introduction section.
We have edited this section accordingly.
- ABHR does not appear to be effective disinfectant for nursing students. This is missing from the conclusion.
This could be the true. However, due to discrepancy in the pre-disinfection bacterial concentrations we think its best to be careful when commenting differences in effects between WBHR and ABHR. However, we have edited the conclusion for clarity.
- Conclusion section is weak and has a misleading statement. It needs to be re-written with recommendations.
We have rewritten this section for clarity.

Reviewer 2 Report
This manuscript, entitled “Effect of optimal alcohol-based hand rub among nurse students compared with everyday practice among random adults. Can water-based hand rub combined with a hand dryer be an alternative?”, had compared the disinfection effect of ABHR, WBHR, and hand-dryer on hand bacterial contamination of skilled health workers (nurse students) and random adults.
Please follow the journal instructions for authors: “References must be numbered in order of appearance in the text.”
L63: What is “HCWs” mean? Please add the full name as in line 65.
L130: “bacterial concentration >109 CFU/mL.” I would like to add approximately a number of the CFU/ml.
L141: In the European Standard EN 1500:2013: they evaluate the efficacy of alcohol against E. coli; in the current study, the authors evaluate the efficacy of alcohol, water, and air drier. Why do authors not examine the effect of such agents on gram-positive bacteria and fungi?
L174: “post-samples were taken from dry hands” how many seconds or minutes passed to reach a dry hand status?
Author Response
Reviewer 2:
Comments and Suggestions for Authors
This manuscript, entitled “Effect of optimal alcohol-based hand rub among nurse students compared with everyday practice among random adults. Can water-based hand rub combined with a hand dryer be an alternative?”, had compared the disinfection effect of ABHR, WBHR, and hand-dryer on hand bacterial contamination of skilled health workers (nurse students) and random adults.
- Please follow the journal instructions for authors: “References must be numbered in order of appearance in the text.”
This is corrected.
- L63: What is “HCWs” mean? Please add the full name as in line 65.
Corrected.
- L130: “bacterial concentration >109 CFU/mL.” I would like to add approximately a number of the CFU/ml.
Corrected.
- L141: In the European Standard EN 1500:2013: they evaluate the efficacy of alcohol against E. coli; in the current study, the authors evaluate the efficacy of alcohol, water, and air drier. Why do authors not examine the effect of such agents on gram-positive bacteria and fungi?
We will certainly consider a study that include gram-positive bacteria and fungi. However, this was not the scope of our study this time.
L174: “post-samples were taken from dry hands” how many seconds or minutes passed to reach a dry hand status
This information has been added.
Reviewer 2:
Comments and Suggestions for Authors
This manuscript, entitled “Effect of optimal alcohol-based hand rub among nurse students compared with everyday practice among random adults. Can water-based hand rub combined with a hand dryer be an alternative?”, had compared the disinfection effect of ABHR, WBHR, and hand-dryer on hand bacterial contamination of skilled health workers (nurse students) and random adults.
- Please follow the journal instructions for authors: “References must be numbered in order of appearance in the text.”
This is corrected.
- L63: What is “HCWs” mean? Please add the full name as in line 65.
Corrected.
- L130: “bacterial concentration >109 CFU/mL.” I would like to add approximately a number of the CFU/ml.
Corrected.
- L141: In the European Standard EN 1500:2013: they evaluate the efficacy of alcohol against E. coli; in the current study, the authors evaluate the efficacy of alcohol, water, and air drier. Why do authors not examine the effect of such agents on gram-positive bacteria and fungi?
We will certainly consider a study that include gram-positive bacteria and fungi. However, this was not the scope of our study this time.
L174: “post-samples were taken from dry hands” how many seconds or minutes passed to reach a dry hand status
This information has been added.
Reviewer 3 Report
Dear authors ,
your paper is very interesting and addresses an important and ongoing current issue after Covid-19 pandemic, however in my opinion it is necessary to improve all the cited references and the conclusions adding a take home message.
Kind regards
Author Response
Reviewer 3
Dear authors,
- your paper is very interesting and addresses an important and ongoing current issue after Covid-19 pandemic, however in my opinion it is necessary to improve all the cited references and the conclusions adding a take home message.
We have edited the conclusions regarding a clearer take home message.
We have also made changes in the reference list, and omitted a couple of the references.
Round 2
Reviewer 1 Report
Dear authors
Thank you for revising the manuscript and incorporating the suggestions. I have no further suggestions. Well done!